# Preservation of three-dimensional anatomy in phosphatized fossil arthropods enriches evolutionary inference

**Achim H Schwermann[1]\*, Tomy dos Santos Rolo[2], Michael S Caterino[3], Günter Bechly[4], Heiko Schmied[5], Tilo Baumbach[2,6], Thomas van de Kamp[2,6]\***

[1] Steinmann Institute for Geology, Mineralogy and Paleontology, University of Bonn, Bonn, Germany; [2] ANKA/Institute for Photon Science and Synchrotron Radiation, Karlsruhe Institute of Technology, Eggenstein-Leopoldshafen, Germany; [3] Department of Agricultural and Environmental Sciences, Clemson University, Clemson, United States; [4] State Museum of Natural History Stuttgart, Stuttgart, Germany; [5] Institute of Crop Science and Resource Conservation, University of Bonn, Bonn, Germany; [6] Laboratory for Applications of Synchrotron Radiation, Karlsruhe Institute of Technology, Karlsruhe, Germany

**Abstract** External and internal morphological characters of extant and fossil organisms are crucial to establishing their systematic position, ecological role and evolutionary trends. The lack of internal characters and soft-tissue preservation in many arthropod fossils, however, impedes comprehensive phylogenetic analyses and species descriptions according to taxonomic standards for Recent organisms. We found well-preserved three-dimensional anatomy in mineralized arthropods from Paleogene fissure fillings and demonstrate the value of these fossils by utilizing digitally reconstructed anatomical structure of a hister beetle. The new anatomical data facilitate a refinement of the species diagnosis and allowed us to reject a previous hypothesis of close phylogenetic relationship to an extant congeneric species. Our findings suggest that mineralized fossils, even those of macroscopically poor preservation, constitute a rich but yet largely unexploited source of anatomical data for fossil arthropods.

\*For correspondence: achim. schwermann@uni-bonn.de (AHS); thomas.vandekamp@kit.edu (TvdK)

**Competing interests:** The authors declare that no competing interests exist.

## Introduction

An organism's morphology represents a complex solution to myriad ecological and environmental challenges it and its ancestors have confronted over evolutionary time. Inferring a comprehensive evolutionary history of a lineage requires consideration of a wide range of morphological features, and how they may have been shaped by selection, drift, and developmental constraints. While external characters predominate in ecomorphological and systematic studies, internal characters also play critical roles (*Perreau and Tafforeau, 2011*). In fossil specimens, however, these characters are usually not preserved or difficult to access (*Siveter et al., 2007*). While combined phylogenetic analyses of extant species frequently utilize internal anatomy, analyses including fossil taxa are generally limited to external characters. Moreover, it is often difficult to distinguish whether unobserved morphological characters were originally absent or lost due to taphonomic processes, potentially leading to misinterpretations of character evolution and erroneous phylogenetic placements (*Sansom, 2015*).

Several types of preservation or certain combinations of them are known for arthropod fossils. These are adpressions (compressions or impressions) (*Wedmann et al., 2007*; *2011*), casts, voids,

**eLife digest** Fossils are the preserved remains of animals, plants or other organisms. The most highly prized fossils are those that retain their original three-dimensional shape and provide details needed to identify what species it represents, and what its closest living relatives might be. However, even fossils with the most beautifully preserved external anatomy can lack the internal structures that also help to identify its evolutionary history. "Mineralized" fossils are particularly useful for researchers as they form in a process that helps to preserve the internal anatomy of the organism. This type of fossil forms when mineral-laden water surrounds an organism's body so that the minerals are deposited in its cells and turn soft tissues to stone.

In the 1940s, Swiss scientist Eduard Handschin used eight mineralized fossil specimens to describe a 25-40 million-year-old beetle species called *Onthophilus intermedius*. On the basis of the external anatomy of the two best-preserved specimens, Handschin claimed this species was distinct from, but closely related to a beetle species called *O. striatus* that is found in Europe today.

Since then, fossil examination methods have greatly advanced and include three-dimensional X-ray based imaging techniques that reveal the internal structures of a fossil while leaving it intact. One such technique is called X-ray computed tomography, in which numerous X-ray images of a solid object are taken from different angles. These images are then reassembled using computer software to create a virtual three-dimensional model of the object.

Here, Schwermann et al. used this X-ray technique to re-examine the beetle fossils originally reported by Handschin. This analysis revealed many new details of these specimens' external and internal anatomies, including their gut, genitals and airways. These new insights place *Onthophilus intermedius* into a different evolutionary lineage to *O. striatus*. They also suggest that mineralized fossils could provide a rich source of data for studies on fossil insects and other arthropods, even if they appear to be poorly preserved on the outside.

embeddings, mineral replications, charcoalified remains, or inclusions in amber (*Grimaldi et al., 1994*; *Martínez-Delclòs et al., 2004*; *Grimaldi and Engel, 2005*; *Dunlop and Garwood 2014*; *Penney and Jepson, 2014*). Amber inclusions are famous for exquisitely preserving three-dimensional external shape and sometimes internal characters (*Perreau and Tafforeau, 2011*). Three-dimensional arthropods may also be preserved within concretions (e.g. in siderite nodules [*Nitecki, 1979*; *Garwood et al., 2009*]), calcareous incrustations (e.g. in travertine [*Rosendahl et al., 2013*]), encapsulations in minerals (e.g. in onyx-marble [*Pierce, 1951*], chert [*Anderson and Trewin, 2003*], or gypsum crystals [*Schlüter et al., 2003*]), and mineral replications (e.g. as calcite [*McCobb et al., 1998*], silica [*Miller and Lubkin, 2001*], goethite [*Grimaldi, 2009*; *Barling et al., 2014*], pyrite [*Grimaldi and Engel, 2005*], or phosphate [*Duncan and Briggs, 1996*; *Hellmund and Hellmund, 1996*; *Waloszek, 2003*]). Some of these preservation types have revealed surprisingly detailed insights into the internal and soft tissue anatomy of several arthropods, for instance from several Paleozoic marine deposits (e.g. *Siveter et al., 2007*; *2013*; *2014*; *Ma et al., 2014*; *Cong et al., 2014*; *Edgecombe et al., 2015*). For insects, e.g. eyes (*Duncan and Briggs, 1996*) and muscle fibers (*Grimaldi, 2009*) have been reported.

Abundant arthropod fossils preserved by mineralization of calcium phosphate are known from the Oligocene fissure fillings of Ronheim (*Hellmund and Hellmund, 1996*), the Late Oligocene/Early Miocene limestones of Riversleigh (QLD, Australia) (*Duncan and Briggs, 1996*) and from Paleogene deposits at Quercy (south-central France) (*Filhol, 1877*; *Gervais, 1877*; *Flach, 1890*; *Thévenin, 1903*; *Handschin, 1944*). These localities have long been famous for their rich vertebrate fossils as well (e.g. *Legendre et al., 1997*; *Laloy et al., 2013*). The arthropod fossils of Quercy were documented by Swiss entomologist *Eduard Handschin (1944)*. He described the hister beetle *Onthophilus intermedius* (Coleoptera: Histeridae) from eight specimens, and considered it distinct but closely related to the extant European species *O. striatus* (Forster, 1771). The description, however, was vague and based mainly on the external morphology of the two best-preserved specimens (*Handschin, 1944*).

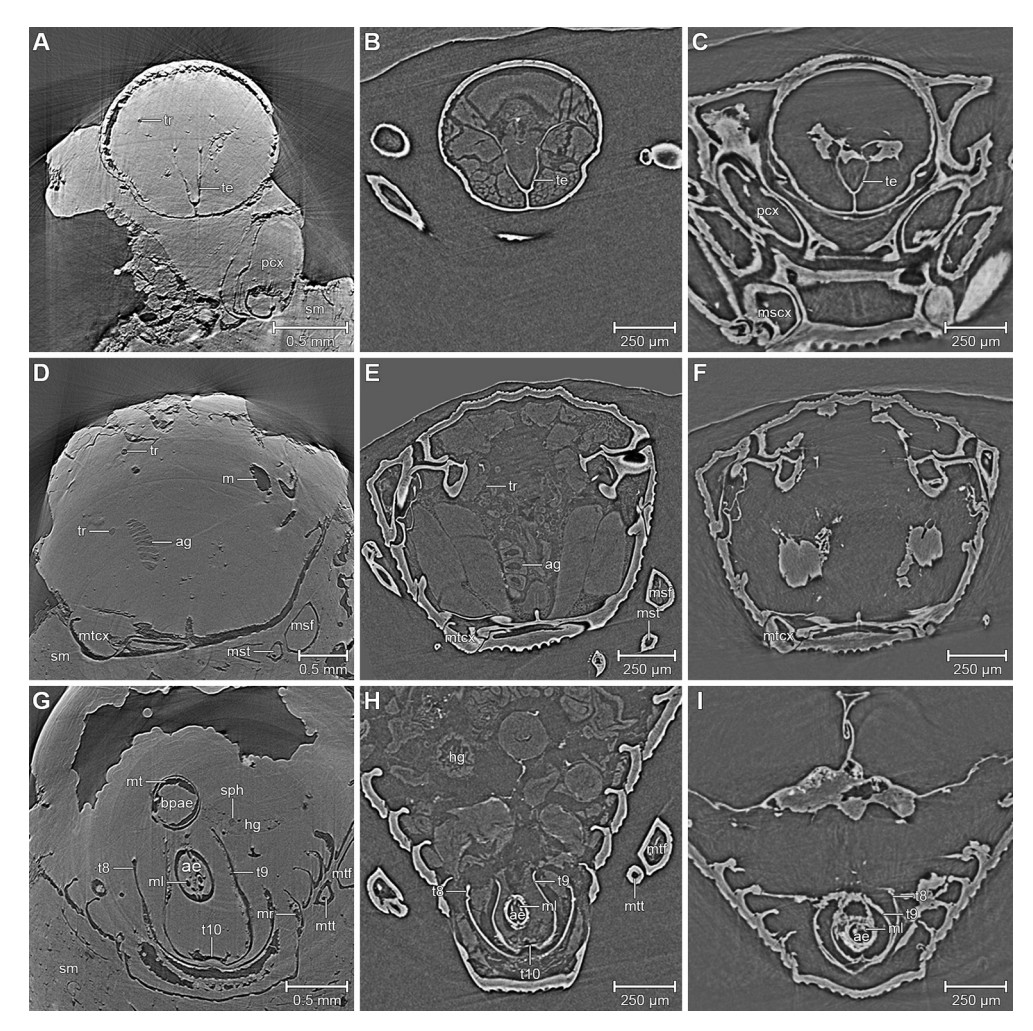

**Figure 1.** Comparison between the fossil *Onthophilus intermedius* (**A**, **D**, **G**) and EtOH-fixed (**B**, **E**, **H**) and air-dried (**C**, **F** , **I**) specimens of *O. striatus*. Slices of tomographic volumes showing head region (**A**–**C**), thorax (**D**–**F**) and abdomen (**G**–**I**). ae = aedeagus; ag = accessory gland; bpae = basal part of aedeagus; hg = hindgut; m = musculature; ml = median lobe; mr = muscles remnants; mscx = mesocoxa; msf = mesofemur; mst = mesotibia; mt = muscle tissue; mtcx = metacoxa; mtf = metafemur; mtt = metatibia; pcx = procoxa; sph = spherical particle; sm = stony matrix; t8 = 8th abdominal tergite; t9 = 9th abdominal tergite; t10 = 10th abdominal tergite; te = tentorium; tr = trachea.

X-ray microtomography has become established for the detailed examination of both extant (e.g. *Betz et al., 2007*; *Bosselaers et al., 2010*; *van de Kamp et al., 2011*; *2014*; *2015*; *Brehm et al., 2015*; *Sombke et al., 2015*) and extinct (*Sutton, 2008*; *Sutton et al., 2014*) arthropods, including fossils preserved in amber (*Lak et al., 2009*; *Pohl et al., 2010*; *Soriano et al., 2010*; *Perreau and Tafforeau, 2011*; *Riedel et al., 2012*). We explored the application of this technique to mineralized fossils by re-examination of Handschin's specimens of *Onthophilus intermedius*. To ensure a direct morphological comparison, we performed tomographic scans (*Figure 1*) of ethanol-fixed and air-dried *O. striatus* using the same experimental setup. Furthermore we tested the hypothesis that the two are closely related with a global phylogenetic analysis of *Onthophilus* Leach, 1817.

**Table 1.** Notes on the fossil *Onthophilus intermedius* specimens from Quercy and their preservation.

| ID | Internal structures preserved | Notes |
|---|---|---|
| F1951 | some sclerites (incl. coxa-trochanteral joints) and tracheae | the only specimen depicted by *Handschin (1944)*; but not explicitly designated as holotype |
| F1992 | some sclerites and small tracheae | head, prothorax missing |
| F1993 | some sclerites (incl. coxa-trochanteral joints) | head, pygidia missing; elytra partly abraded |
| F1994 | most sclerites, muscle parts, tracheae, parts of alimentary system, large parts of male genitals | the only specimen of the collection that is ventrally encrusted by a stone matrix |
| F1995 | some sclerites, parts of male genitals | head present; abdomen deeply abraded dorsally |
| F1996 | some sclerites | head, prothorax missing |
| F1997 | some sclerites, remains of muscles below the elytra | head, prothorax partly abraded |
| F1998 | some sclerites (incl. coxa-trochanteral joints), parts of female genitalia | head, prothorax partly abraded |

## Results and discussion

We found internal characters in all fossils (*Table 1*). Three specimens show remains of inner organs, especially of the sclerotized genitalia, allowing their identification as two males and one female. The outer surfaces of most specimens appear smooth (*Figure 2*); the distinct punctuation found in extant *Onthophilus* species (*Kovarik and Caterino, 2005*) is faint.

The specimen F1994 (*Figures 1A,D,G*, *2*, *3*, *Supplementary file 1*) differs from all other samples by the presence of a stony matrix, covering the ventral part of the beetle. Its dorsal part and head are exposed; the elytra are missing and were probably detached before embedding. The exposed surface is partly eroded, especially in the anterior region of the head, and no appendages are visible from the outside. The matrix, however, concealed the best-preserved fossil from the collection, which we examine here in detail.

The ventral portion of the beetle covered by the matrix reveals an extraordinary preservation of exoskeletal fine structures and internal anatomy (*Figures 3*, *4* and *5*; *Supplementary file 1*). While some fractions of the cuticle appear to be mineralized, the latter is mostly represented by air-filled spaces in the fossil (*Figure 1A,D,G*). The surface of the exoskeleton is preserved as a three-dimensional imprint of remarkable detail; the body sclerites show characteristic punctuation of the genus. The right foreleg is not preserved; the left one is truncated from the trochanter; distal parts of the leg were lost prior to fossilization. The right mid and hind legs are eroded at the edge of the matrix, but their encrusted left counterparts appear complete except for the most distal part of the metafemur of the hind leg that would protrude from the matrix. Moreover, many anatomical characters can be recognized inside the fossil (*Figure 3D*). Apart from internal invaginations of the exoskeleton (e.g. tentorium, furcal arms and metendosternite), large parts of the alimentary canal and tracheal system are visible. The oesophagus appears to be shrunken and is connected to the crop, which is truncated posteriorly. The anterior part of the hindgut is hollow, while the middle part is apparently filled with mineral matrix but well-defined. Conspicuous spherical particles may constitute remnants of gut content (*Figure 1G*). The hindmost part of the gut can be roughly retraced by aggregations of tiny holes inside the mineral matrix. Like in the alimentary canal, some large tracheae appear to be filled with matrix, while others are hollow. Except for the musculature connecting the right pro- and mesofurcal arms (*Figure 1D*), most muscles can only be recognized by remnants at the insertion areas (*Figure 1G*). The genitals are extraordinary well-preserved (*Figure 3F*). While testes and Ductus ejaculatorius could not be recognized, other soft tissues such as the spiral accessory glands and parts of the gland ducts are conspicuous. The genital sclerites, including aedeagus, median lobe, gonopore, tergites 8-10 and sternites 8 & 9 are almost perfectly preserved as imprints.

The remarkable preservation state of the fossil is emphasized when its morphological characters are compared to those of an extant ethanol-fixed specimen of the same genus (*Figures 1*, *3E,F* and *4*). The new anatomical data from this specimen facilitated an extended description of the species according to modern taxonomic standards (Appendix 1).

*Handschin (1944)* hypothesized a close relationship ('particularly striking similarity') between *Onthophilus intermedius* and *O. striatus* based on then-observable external morphology. However,

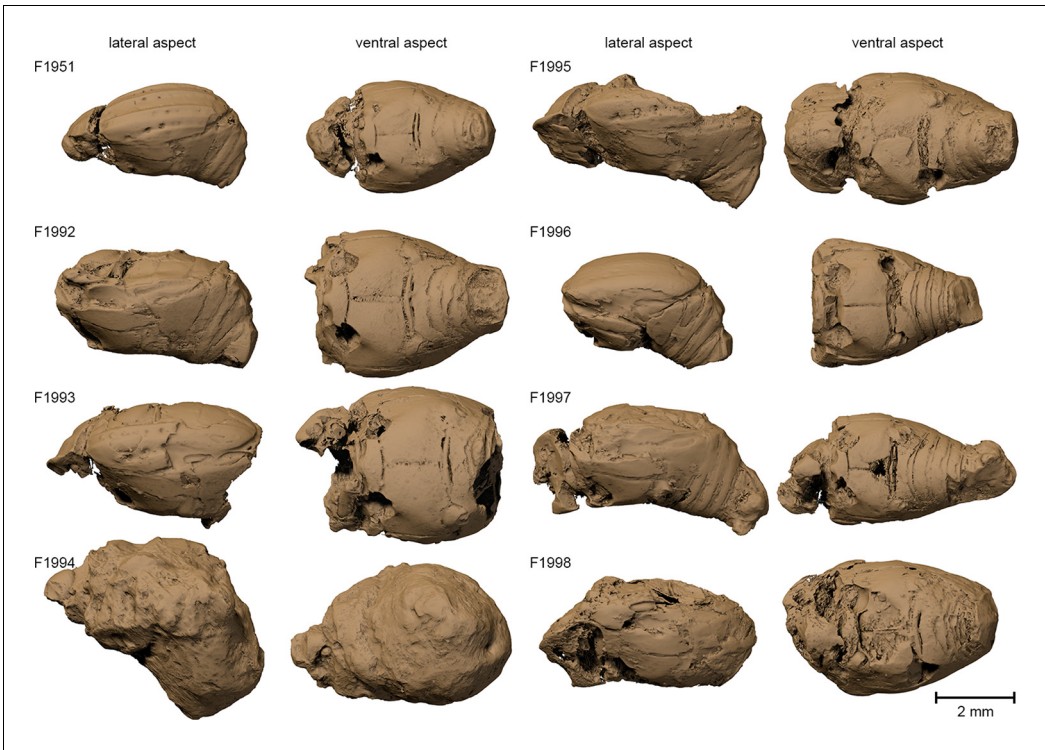

**Figure 2.** Surface renderings of the eight *Onthophilus intermedius* specimens. Note the unique encrustation of F1994.

phylogenetic analysis (Material and methods) of the more diverse character set now accessible places these species in distinct clades. The analysis resulted in 72 most parsimonious trees of length 185 (CI 0.27, RI 0.61). The strict consensus of these trees (*Figure 6*) is well resolved apart from a few rearrangements of some outgroup taxa and within a relatively derived group related to *O. niponensis* Lewis, 1907. *O. intermedius* is part of a trichotomy involving *O. silvae* Lewis, 1884 and a large group of species descended from the common ancestor of *O. giganteus* Heleva, 1978 and *O. niponensis*, though in reweighted trees it is resolved as sister to *O. silvae* alone. In all analyses *O. striatus* is nested within a lineage of Nearctic and far-eastern Palaearctic species, including *O. flavicornis* Lewis, 1884, *O. flohri* Lewis, 1888 and others.

Inclusion of diverse characters revealed by microtomography of *Onthophilus intermedius* yields a well-supported topology and a more comprehensive picture of the biogeographic and morphological history of the group. Of the characters scored for both *O. intermedius* and *O. striatus*, there are seven by which their states differ, three external and four internal. Of these, two external (chars. 29 & 30) and one internal (char. 36) are reconstructed as autapomorphies (*Figure 6*). Only one external synapomorphy (char. 22) separates them, while three of the four genitalic differences (chars. 35, 40, and 41) represent synapomorphies of their respective lineages. Exclusion of internal characters for *O. intermedius* did not affect the topology, but did prevent genitalic characters from supporting its larger containing clade. Critical diagnostic differences in external morphology, such as mesoventral proportions and pygidial sculpturing, were also revealed by visualization of features previously obscured by matrix.

Based on our examinations we can reconstruct the probable fossilization process of the Quercy *Onthophilus* specimens, which culminates in a partial mineralization of inner organs in combination with the cuticle preserved as voids. An accurate three-dimensional conservation of soft tissues does not occur if the specimens are dried in air (*Figure 1C,F,I*). Therefore, the fixation process must have occurred fast, possibly due to the animal being immediately penetrated and enclosed by phosphate rich water. In arthropods, this type of fossilization is only known from a handful of localities, which are better known for a rich vertebrate fauna (Riversleigh: *Duncan and Briggs, 1996*; Ronheim:

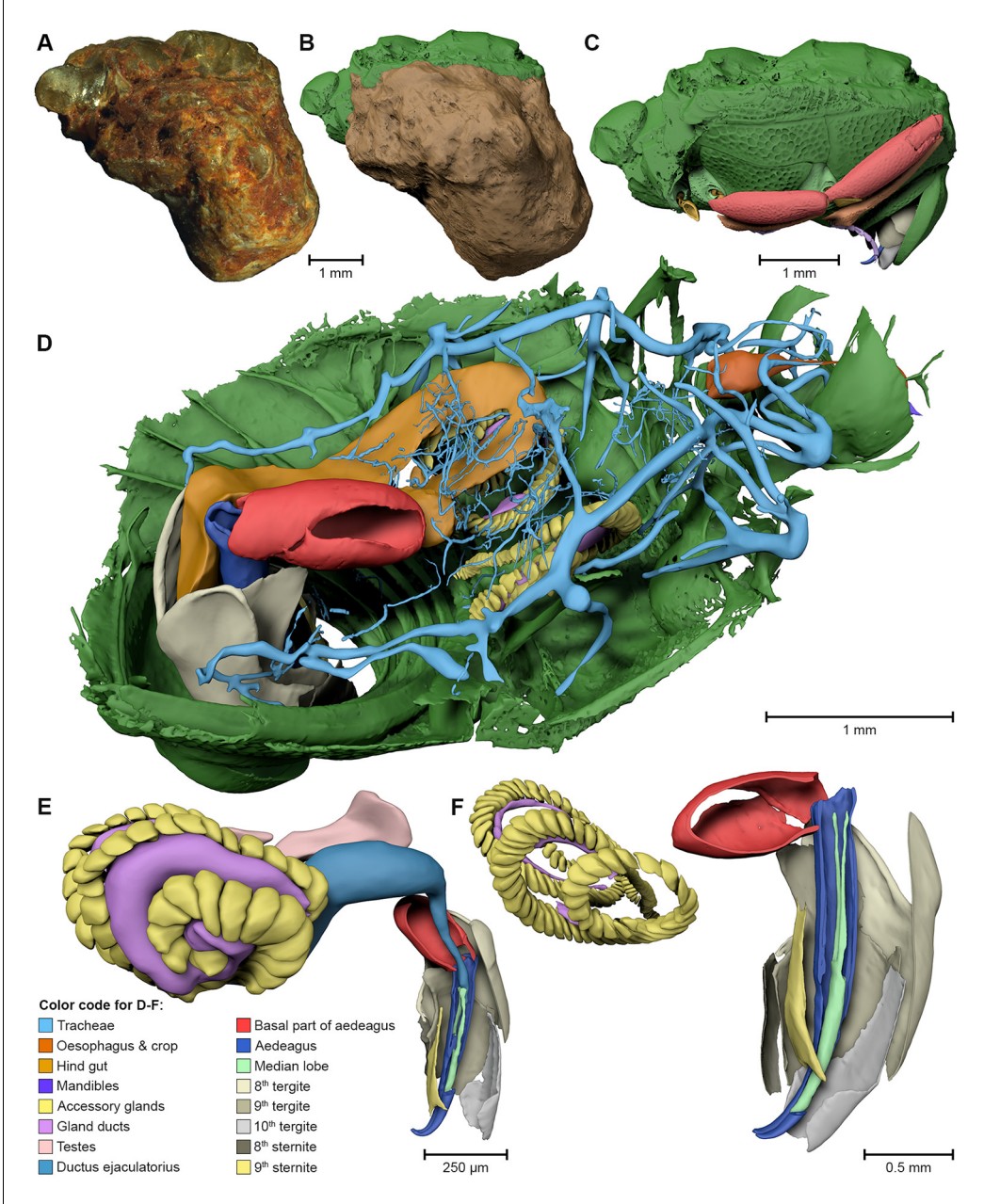

**Figure 3.** Digital reconstruction of the fossil. (**A**) Photograph of *Onthophilus intermedius* (F1994) ventrally embedded in a stony matrix. (**B**) Digital reconstruction showing fossilized beetle (green) and matrix (brown). (**C**) Beetle digitally isolated from the stone, revealing well-preserved morphology hidden by the matrix. (**D**) Perspective view of the fossil showing parts of exoskeleton, tracheal network, alimentary canal and genitals. (**E, F**) Comparison of the male genitals of the extant *O. striatus* (**E**) and the fossil *O. intermedius* (**F**); outer sclerites cut to reveal internal anatomy. See *Supplementary file 1* for an interactive version of the 3D reconstruction.

*Hellmund and Hellmund, 1996*; Quercy: *Handschin, 1944*). Replication of soft tissues by phosphatization may be accomplished over a period of weeks (*Martínez-Delclòs et al., 2004*). Possible sources for high phosphorous concentrations in water circulating through the fissure fill are rocks or abundant phosphate-rich vertebrate bones, which may have been deposited along with them (*Handschin, 1944; Hellmund and Hellmund, 1996*). After encrustation and internal mineralization, the cuticle largely decayed, leaving air-filled spaces. Erosion processes probably removed the outer stony matrix of most specimens, including fragile appendages and the imprint of the outer surface

**Figure 4.** Coxa-trochanteral joints. Comparison of the joints (cut) of the left mid- (**A**, **B**) and hind leg (**C**, **D**) of *Onthophilus striatus* (**A**, **C**) and *O. intermedius* (**B**, **D**), showing coxae (green) and trochanters (yellow).

of the exoskeleton, leaving a mineralized endocast. Thus, the exterior of the fossils merely represents the inner surface of the exoskeleton – the deep grooves (*Figure 2*) actually being inner folds or apophyses. While the smooth dorsal part of F1994 resembles the other fossils in appearance, its ventral surface covered by the mineral matrix shows a distinct surface sculpturing as present in extant species of the genus. In contrast, an artificial 'digital endocast' created from the tomographic data of F1994 (Material and methods) bears a striking resemblance to the other fossils (*Figure 5*), on which Handschin based his original description. Summing up, the Quercy hister beetles represent three-dimensional 'hybrid' fossils, comprising cuticle imprints and mineralized soft tissue, combining to preserve both exoskeletal fine structure and internal anatomical characters.

Fissure filling fossils preserving three-dimensional internal anatomy will help to overcome taphonomic biases in available fossil data (*Allison and Bottjer, 2011*). To date, fossilized insect internal character information has mainly been obtained from well-preserved amber inclusions (e.g. *Pohl, et al., 2010*; *Perreau and Tafforeau, 2011*). However, the origination of amber as tree resin causes a representational bias toward generally arboreal taxa (*Martínez-Delclòs et al., 2004*). The fossil arthropods of Quercy represent an assemblage of taxa more typically associated with forest floor communities (*Handschin, 1944*), as exemplified by *Onthophilus*, typically a predator in various decaying organic materials (*Kovarik and Caterino, 2005*; *Bajerlein et al., 2011*). Such communities are less commonly preserved than those of many other environments (*Kidwell and Flessa, 1996*). Beyond anatomical data on these species, clearer interpretations of evolutionary relationships of these fossils will improve inferences about the evolution of these ecological communities. Thus, reexamination of the Quercy fossils, and likely also of similar mineralized fossils from other localities (which may represent different ecosystems and/or time periods), may provide a highly complementary source of information on the evolutionary history of arthropods.

With regard to the methods employed here, we can offer some guidance on improving future imaging attempts on similar materials. Based on our experience, a fast tomography setup combining filtered polychromatic radiation and an optimized detector system (*dos Santos Rolo et al., 2014*) is well-suited to achieve sufficient image quality in most fossil specimens. Thus, scan duration per tomogram may be reduced to a couple of seconds (Material and methods), facilitating high-through-put screening of large sample numbers in short time.

Our results demonstrate that mineralized arthropod fossils from a sedimentary context may three-dimensionally preserve soft tissue and other internal anatomical characters in remarkable detail, which allows determinations and phylogenetic analyses according to the standards for Recent organisms. Reevaluation of relationships with modern taxa in this extended morphological context will substantially improve estimates of rates and modes of arthropod evolution. This exceptionally detailed preservation may be aided by the presence of a surrounding stony matrix, hinting that encrusted specimens, which therefore were originally considered to be of poor quality, could contain particularly well-preserved external and internal characters. Our findings may trigger the reinvestigation of numerous similar fossils from various localities.

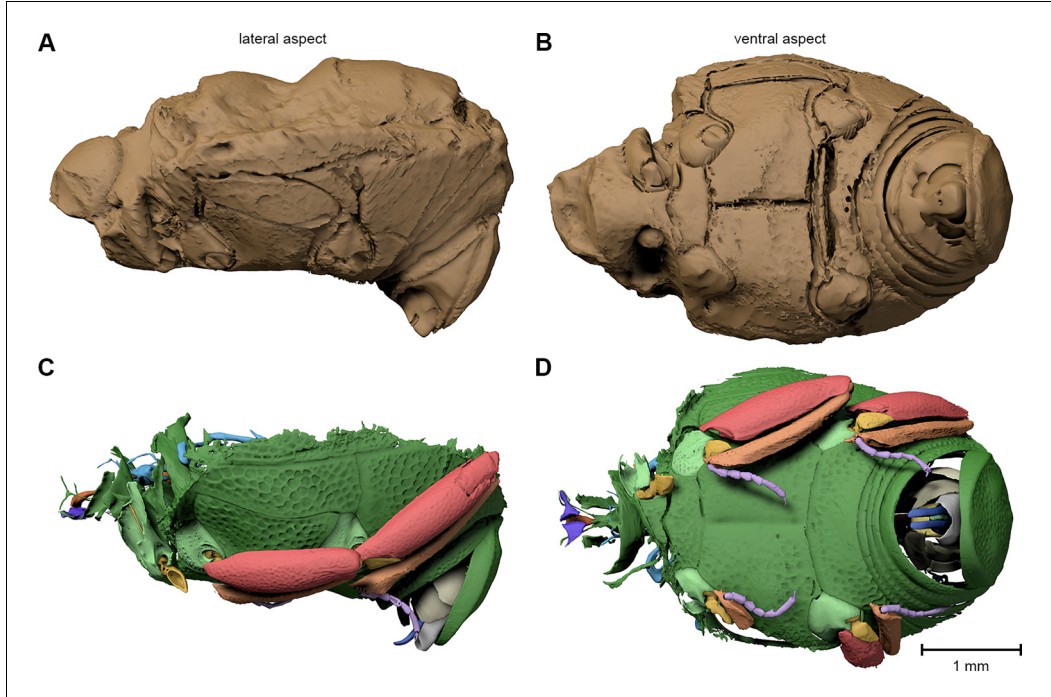

**Figure 5.** Digital endocast of *Onthophilus intermedius* (specimen F1994). A digital endocast (**A**, **B**) artificially created from tomography data resembles the shape of the other fossils (*Figure 2*) much closer than the original surface of the beetle (**C**, **D**) hidden by the stony matrix.

## Materials and methods

### Synchrotron X-ray microtomography

3D X-ray micro-computed tomography scans with synchrotron radiation (μCT) were performed at the TOPO-TOMO beamline (*Rack et al., 2009*) of the ANKA Synchrotron Radiation Facility at Karls-ruhe Institute of Technology (KIT). The measurements consisted of the acquisition of 2500 equiangu-larly spaced radiographic projections of the sample in a range of 180°. The frame rate was set to 150 images per second, resulting in an overall scan duration of 16.67 seconds per sample. The parallel polychromatic X-ray beam produced by a 1.5 T bending magnet was spectrally filtered by 0.2 mm aluminum to obtain a peak at about 15 keV. The sample was placed 20 cm upstream of the detector, which in turn was located about 33 m from the source. The detector consists of a thin, plan-parallel lutetium aluminum garnet single crystal scintillator doped with cerium (LuAG:Ce), optically coupled via a Nikon Nikkor 85/1.4 photo-lens to a pco.dimax camera with a pixel matrix of 2008x2008 pixels. The lens was stopped down to F/4 to remove optical aberrations and to increase its depth of focus, permitting the use of a thicker scintillator to collect a higher fraction of the incident X-ray photons. The magnification of the optical system was adjusted to 3X, yielding an effective X-ray pixel size of 3.66 μm (*dos Santos Rolo et al., 2014*). Tomographic reconstruction was performed with the GPU-accelerated filtered back projection algorithm implemented in the software framework UFO (*Vogelgesang et al., 2012*). Microtomographic image data are deposited in Morph·D·Base (www.morphdbase.de; accession numbers T_vandeKamp_20151216-M-12.1 to T_vandeKamp_20151216-M-22.1).

### 3D reconstructions

3D reconstruction followed the protocol described by *Ruthensteiner and Heß (2008)* and *van de Kamp et al., (2014)*; using Amira (versions 5.5, 6, FEI) and Avizo (version 8.1, FEI) for segmentation of the tomographic volumes and CINEMA 4D R15 (Maxon Computer GmbH) for assembly of com-ponents and rendering of figures. The 'digital endocast' (*Figure 5*) was created from the

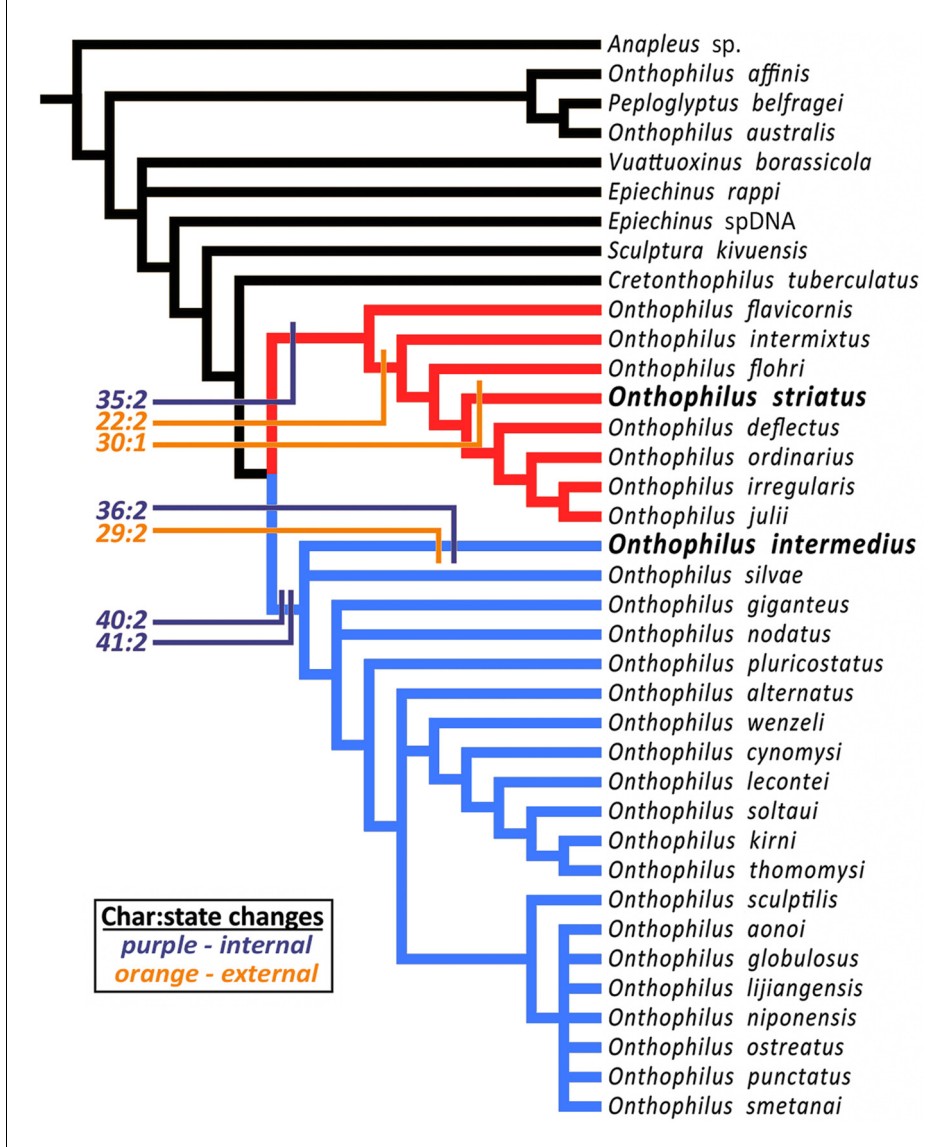

**Figure 6.** Strict consensus tree. The analysis places *Onthophilus striatus* within a lineage of Nearctic and far-eastern Palaearctic species (red), while *O. intermedius* is a member of a separate Holarctic lineage (blue). Four internal (purple) and three external (orange) unambiguous synapomorphies supporting their respective placements are mapped onto the cladogram - *Onthophilus striatus* group: Character 22:2, mesoventrite wide and short; 30:1, pygidial median carina absent; 35:2, tegmen of aedeagus abruptly downturned apically.
*O. intermedius* group: 29:2, pygidium laterally impunctate; 36:2, tegmen of aedeagus abruptly narrowing apically; 40:2, lateral halves of eighth sternite large and nearly meeting at midline; 41:2, stem of spiculum gastrale broad throughout its length.

tomographic stack of specimen F1994 by segmenting solely the dorsal stony matrix, ventrally confined by the inner impression of the beetle's cuticle.

The number of surface polygons was reduced to 10% of its original value in CINEMA 4D: the raw mesh of F1994 contains approx. 30 million polygons, the reduced version (*Figure 3D*) ca. 3 million. Segmentation artifacts were carefully removed using the sculpting tools of the software. For the interactive 3D model (*Supplementary file 1*), the polygon count was further reduced to 800,000 (without the stony matrix); the digital mesh was imported into Deep Exploration (version 6; Right Hemisphere), saved as Universal 3D file (U3D) and embedded into a PDF document with Adobe® Acrobat® 9 Pro Extended.

## Phylogenetic analysis of *Onthophilus intermedius*

Our phylogenetic analysis was performed to test *Handschin's (1944)* hypothesis of a close relationship of *Onthophilus* intermedius to the extant and sympatric *O. striatus*. Although his hypothesis was not presented in strictly phylogenetic terms ('particularly striking similarity'; our translation), the suggestion is of a direct lineal relationship between these heterochronic species. This would be revealed in a cladistic analysis as a sister group relationship between them. Thus, the hypothesis would be rejected by any resolution in which *O. intermedius* and *O. striatus* were not found to be sister species. We compiled a character set comprising 41 characters (*Source code 1*) of internal and external morphology visible in one or more specimens of *O. intermedius*, as visualized following X-ray microtomography. We scored these characters for a set of 29 of the 39 currently described species in the genus *Onthophilus* (*Mazur, 2011*), as well as seven outgroup Onthophilinae (including the recently described Cretaceous *Cretonthophilus tuberculatus* (*Caterino et al., 2015*). Most were scored from direct examination of specimens. However, some taxa were scored from illustrations and descriptions in the literature (*Reichardt, 1941*; *Helava and Howden, 1977*; *Helava, 1978*; *Ôhara and Nakane, 1986*; *Ôhara, 1989*; *Howden and Laplante, 2003*).

## Characters and states

1. Sutures separating antennomeres of antennal club: 1, distinct; 2, indistinct.
2. Position of antennal insertion: 1, at upper edge of eye; 2, in front of middle of eye.
3. Proximity of antennal fovea and eye: 1, antennal fovea in contact with inner edge of eye; 2, separated by cuticular ridge from eye.
4. Median frontal carina: 1, absent; 2, present.
5. V-shaped lateral frontal carinae: 1, absent; 2, present.
6. Labral setae: 1, bisetose; 2, plurisetose (due to secondary setae).
7. Number of pronotal carinae: 1, zero; 2, two; 3. four; 4, six.
8. Form of outer pronotal carina: 1, absent; 2, excavate along inner edge; 3, raised to form a simple carina.
9. Completeness of outer pronotal carinae: 1, complete; 2, anteriorly abbreviated; 3, interrupted; 4, absent.
10. Completeness of median pronotal carinae: 1, complete; 2, abbreviated; 3, interrupted; 4, absent.
11. Consistency of strength of pronotal carinae: 1. all pronotal carinae equal strength; 2, pronotal carinae alternating in strength
12. Pronotal sculpturing: 1, ground punctation absent; 2, simply punctate (finely or deeply); 3, surface reticulo-strigose (punctures elongated and dense).
13. Lateral margin of pronotum: 1, without dense border of punctures along margin; 2, deeply punctate along inner edge of lateral margin.
14. Longitudinal elytral carinae: 1, absent; 2, present.
15. Evenness of elytral carinae: 1, elytral carinae similar in height; 2, alternating in height.
16. Completeness of elytral carinae: 1, All complete; 2, One or more carinae interrupted along its length.
17. Basal elytral foveae (between costae 2 & 4, sensu *Helava (1978)*): 1, without deep basal foveae; 2, with deep basal foveae.
18. Foveae of elytral interstriae: 1, absent; 2, weak; 3, strong.
19. Basal emargination of prosternal keel: 1, not emarginate, truncate or projecting; 2, narrowly, subacutely emarginate; 3, broadly, more obtusely emarginate.
20. Lateral notch of prosternal lobe: 1, without lateral notch; 2, with lateral notch.
21. Spination of protibia: 1, not densely spinose; 2, densely spinose.
22. Proportions of mesoventrite: 1, nearly half as long as wide (ie. length/width ratio ~0.5); 2, wide and short (length/width ratio >0.5).
23. Postmesocoxal stria of metaventrite: 1, absent (or totally obscured by punctures); 2, present.
24. Punctation of metaventral disk: 1, uniform; 2, with discrete impunctate areas on either side of midline.
25. Spination of outer margin of mesotibia: 1, absent; 2, present.
26. Apical lateral spine of mesotibia: 1, absent or weakly produced, not disrupting outer margin of tibia; 2, Well developed, tibial apex produced.
27. Median carina of propygidium: 1, absent; 2, present.

28. Lateral carinae of propygidium: 1, absent; 2, present.
29. Punctation of pygidium: 1, uniform; 2, with discrete impunctate areas on either side of midline.
30. Median longitudinal carina of pygidium: 1, absent; 2, present, single; 3, present, doubled.
31. Transverse carina of pygidium: 1, absent; 2, present.
32. Basal piece, closure: 1, open, not forming a closed cylinder; 2, forming a complete, closed cylinder; 3, fused with tegmen (some *Epiechinus* only).
33. Basal piece, length relative to tegmen: 1, long, nearly half length of tegmen; 2, much less than half length of tegmen.
34. Tegmen midline division: 1, divided along entire midline to base; 2, fused along >1/4 of its length.
35. Tegmen, apical curvature: 1, evenly curved to tip; 2, abruptly downturned at apex.
36. Tegmen, height (as seen in lateral aspect): 1, evenly narrowing; 2, abruptly narrowing near midpoint.
37. Tegmen, relative widths along length: 1, widest in basal half; 2, parallel-sided or widest in apical half.
38. Point of median lobe extrusion (following *Helava (1978)*): 1, near dorsal apex; 2, ventrally, subapical.
39. Tegmen, apices: 1, apices convergent; 2, apices parallel (approximate or separate); 3, apices divergent
40. Development of 8th sternite: 1. lateral halves reduced, broadly separated; 2. halves more substantial, approaching or meeting at midline.
41. Stem of 9th sternite (spiculum gastrale): 1, stem narrow, abruptly widened to apex; 2, stem broad, weakly widened to apex.

Data were analyzed under parsimony using PAUP* 4.0a144 (*Swofford, 2002*), using a heuristic search with 1000 random addition sequence replicates. Characters were all treated as unordered. We examined the effects of character reweighting (by rescaled consistency indices), and exclusion of various character subsets (internal vs. external). Character transitions were mapped using Mesquite v. 3.03 (*Maddison and Maddison, 2015*). The tree was rooted with either *Anapleus* (Dendrophilinae: Anapleini), considered to exhibit plesiomorphic states in many higher level histerid characters (*Caterino and Vogler, 2002*), or *Cretonthophilus*, a recently described taxon from Cretaceous Burmese amber representing the oldest known Onthophiline histerid (*Caterino et al., 2015*) (*Source code 1*).

## Acknowledgements

We are most grateful to W Etter and O Schmidt (Natural History Museum of Basel) for loaning us the fossils and to D Bajerlein (Adam Mickiewicz University in Poznań) for providing fixed and dried specimens of *Onthophilus striatus*. We acknowledge B Mähler (Steinmann Institute, Bonn) for triggering this project and thank T Faragó (KIT, Eggenstein-Leopoldshafen) for his help with the reconstruction of tomographic volumes. We also thank T Hörnschemeyer (University of Göttingen) for his help at an early stage of this project and S Legendre (French National Centre for Scientific Research, Paris) for the explanation of the historical Quercy collections. The ANKA Synchrotron Radiation Facility is acknowledged for providing beamtime.

## Additional information

### Funding

| Funder | Grant reference number | Author |
|--------|------------------------|--------|
| Bundesministerium für Bildung und Forschung | 05K10CKB | Tomy dos Santos Rolo Thomas van de Kamp |
| Bundesministerium für Bildung und Forschung | 05K12CK2 | Tomy dos Santos Rolo Thomas van de Kamp |

The funders had no role in study design, data collection and interpretation, or the decision to submit the work for publication.

## Author contributions
AHS, TvdK, Conception and design, Acquisition of data, Analysis and interpretation of data, Drafting or revising the article; TdSR, Acquisition of data, Analysis and interpretation of data, Drafting or revising the article; MSC, GB, Analysis and interpretation of data, Drafting or revising the article; HS, Conception and design, Drafting or revising the article; TB, Acquisition of data, Drafting or revising the article

## Author ORCIDs
Thomas van de Kamp, http://orcid.org/0000-0001-7390-1318

## Additional files

### Supplementary files
• Supplementary file 1. Interactive 3D reconstruction of *Onthophilus intermedius* specimen F1994. Click on the figure to start interactive 3D view; switch between views by using the menu (Adobe Reader 8.1 or higher required).

• Source code 1. Nexus code for character matrix.

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

## Appendix 1

# Redescription of *Onthophilus intermedius* Handschin, 1944

*Figures 2–5*, *Appendix figure 1* and *Supplementary file 1*

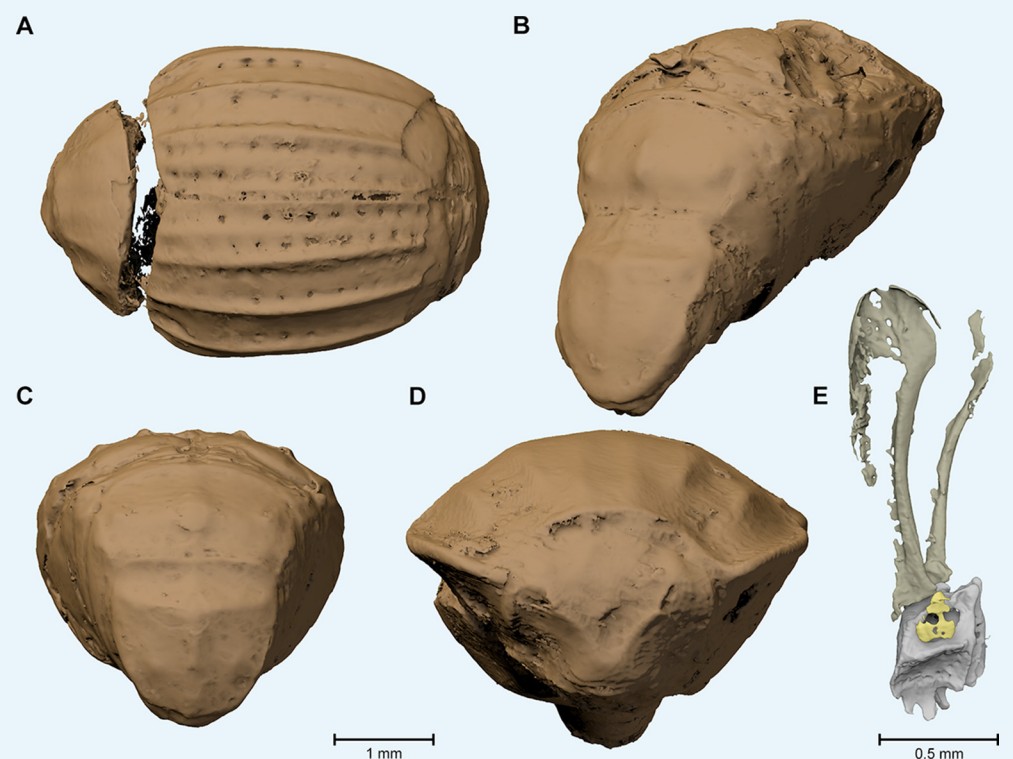

**Appendix figure 1.** Morphological characters visible in specimens other than F1994. (**A**) Dorsal view of F1951, showing elytral carinae and foveae. (**B**) Posterolateral view of F1997, showing propygidial and pygidial carinae. (**C**) Posterior view of F1951. (**D**) Anterior view of F1995. (**E**) Valvifer and coxite of female ovipositor.

## Type locality

"Larnagol (Quercy) 1902. Coll. Rossignol" (*Handschin, 1944*). Fissure fillings from Larnagol are not known. It seems, then, that this is a general reference to the fossils that were collected by Rossignol, residing in Larnagol.

## Type material

Lectotype male, here designated (housed in Natural History Museum of Basel): Specimen F1994, though largely encased in stone matrix, uniquely preserves external and internal morphology suitable for species diagnosis. *Handschin (1944)* explicitly based his description on the two best preserved specimens out of eight, without identifying them or selecting any one as a primary type. We here specify a lectotype due to the highly variable state of preservation of the material available, and considerable possibility of misinterpretation of what are mostly endocasts.

## Other material

Paralectotypes: male (F1995); female (F1998); undetermined sex (F1951, F1992, F1993, F1996, F1997).

## Description

Length: 4.6 mm, width: 2.8 mm; body elongate oval, *distinctly costate on dorsal surfaces; head with convergent frontal carinae (**Appendix figure 1**). Pronotum with six uninterrupted costae; outer and lateral costae (PC1 & PC2 sensu **Helava (1978)** slightly abbreviated, obsolete in anterior one-third; median costae (PC3) complete (**Appendix figure 1D**); lateral pronotal margin slightly elevated, strongly arcuately narrowed from base to apex. Elytra each with three strong, complete dorsal costae (ISC, EC2, & EC4 sensu **Helava (1978)**), EC1, EC3, and EC5 only weakly developed (**Appendix figure 1A**); striae deeply punctured along their lengths. Propygidium about twice as wide as midline length, depressed along anterior margin; disk with distinct median and lateral carinae (PMC and PLC sensu **Helava (1978)**), median carina most strongly produced just behind middle, rapidly diminshing anteriorly and posteriorly; lateral carinae weaker and short, little more than lateral tubercles (**Appendix figure 1B,C**)*; Pygidium slightly longer than wide, with distinct longitudinal and transverse carinae (LC and TC *sensu* **Helava (1978)**), the longitudinal carina varied in strength, *appearing more complete in endocasts*; transverse carina complete, slightly expanded at lateral extremes; pygidial disk conspicuously punctate, with punctures slightly smaller distad, with two small impunctate areas on either side of midline. Prosternal keel emarginate at base (**Appendix figure 1D**); *prosternal lobe short; antennal cavities present in anterior corners of hypomera*. Mesoventrite approximately 1.25x as wide as midline length, subacutely projecting at anterior midpoint, uniformly punctate. Metaventrite rather deeply depressed along midline, rather shallowly and sparsely punctate medially, with larger and coarser punctures posterad and laterad, becoming densely and coarsely punctate at sides; metepisternum and metepimeron similarly coarsely and uniformly punctate. First visible abdominal ventrite weakly punctate at middle, more coarsely punctate near metacoxae; visible ventrites 2-4 with single series of rather small punctures, plus all abdominal ventrites crenulately punctate along posterior margins. Meso- and metatrochanters produced at apices; meso- and metafemora punctate on outer surfaces, metafemur more distinctly elongate; mesotibia weakly curved inward, with weak longitudinal carina on outer surface, stronger carina along posterior margin, apex oblique, weakly hooked at inner apex, not obviously produced at outer apex; metatibia more elongate, more weakly curved, and more distinctly produced at outer apex. Tarsi with basal tarsomeres about 1.5x as long as tarsomeres 2-4, apicalmost tarsomeres about twice as long as tarsomeres 2-4.

Male genitalia (*Figure 3F*): T8 basally deeply emarginate, widened slightly from base to middle, apical margin weakly emarginate; S8 divided medially, sides slightly separated, subtriangular, articulating at basal corners with ventrolateral process of T8; T9 deeply emarginate dorsally, with broad basal apodemes; S9 (spiculum gastrale) with base about one-half width of apex, evenly widened along its length; T10 not subdivided, about twice as broad as long, widest at midpoint, more strongly narrowed to base than to apex, apex shallowly emarginate; basal piece of aedeagus slightly bulbous, about half as long as tegmen, open on left side (not a continuous cylinder), basal foramen opening left; tegmen narrow, subparallel sided in basal half, narrowed slightly toward apex, divided along midline in apical third, apices separate, parallel to tips, apical half curving gradually downward with apices ultimately perpendicular to main tegmen axis; median lobe nearly as long as tegmen, with basal apodemes constituting about half its length, probably extruded dorsally (following *Helava (1978)*).

Female (*Appendix figure 1E*): Valvifers broad basally, narrowing slightly beyond midpoint, then expanded to articulation with coxites; coxites rather short, scoop-shaped, quadridentate, with second inner tooth most strongly produced, coarsely punctured on outer and inner surfaces; gonostyle present on inner surface between second and third apical teeth.

## Remarks

In the description above, we denote characters based only on endocast specimens in italics, because it is not possible to know exactly how these manifested on the external surface. Given the characters available, this species can be distinguished from other *Onthophilus* externally by the impunctate lateral areas on the pygidium, and internally by a tegmen that curves rather gradually from the midpoint to the apex, with its apices parallel but slightly separated over about the apical one-third. Phylogenetic analyses place it in a rather isolated position in the genus, without close relatives among extant species.

