## [Decision Letter]

Thank you for submitting your work entitled "Preservation of three-dimensional anatomy in phosphatized fossil arthropods enriches evolutionary inference" for consideration by *eLife*. Your article has been reviewed by two peer reviewers, one of whom, Russel Garwood, has agreed to reveal his identity. The evaluation has been overseen by Diethard Tautz as the Senior Editor.

The reviewers have discussed the reviews with one another and the editor has drafted this decision to help you prepare a revised submission.

Summary:

The authors show that the internal preservation of the Quercy fossils is unexpectedly excellent, and the authors reveal this with technical skill. The level of preservation is incredible for a non-amber fossil. The preservation of soft anatomy such as an insect's tracheal system makes the study just the kind of thing that should have a large impact in palaeontology. Further, there is actually more behind this manuscript than a technical/techniques paper, and in fact the phylogenetic finding that the Quercy fossil histerid species is most closely related to a North American species is interesting (should mention this in the Abstract). Surely, this biogeographic finding would not have been possible without such remarkable preservation. Regarding the mode of preservation, it is also quite likely that internal decomposition of the original insect was quickly halted because of the relatively antiseptic nature of the phosphate-saturated water that infiltrated the body.

Essential revisions:

One of the reviewers (Russel Garwood) provides direct comments on text passages that should be responded to (appended below). Several of them refer to providing more citations for the general technical background of the study.

Please clarify the following sentences in light of the accompanying comments:

Abstract:

1) “Internal characters in fossil arthropods are crucial to establishing systematic position, ecological role and evolutionary trends.”

I believe this is an oversimplification of a complex issue – see comments below.

Introduction:

2) “Internal characters are crucial for assessing the systematic placement and ecological role of organisms and provide essential information for reliable evolutionary inference. In many arthropods, they include critical systematic characters (Perreau and Tafforeau, 2011).”

This statement is a simplification of what is in fact a complex and under-researched area. Internal characters in fossils are just a form of missing data, and are no more – or less – important, than external data. I suggest the authors check out the following paper: Sansom, R. S. (2014). Bias and sensitivity in the placement of fossil taxa resulting from interpretations of missing data. Systematic biology, syu093.

It’s relatively safe to say that more data is useful in a cladistic phylogenetic context, but (as Sansom 2015 demonstrates) – what is more important when dealing with missing data is to correctly code data as such, rather than introducing false absences (which is the biggest risk when dealing with fossils). Nobody has, to my knowledge, published any work on whether the loss of internal characters leads to biases in phylogenies. This is where it would be a significant issue. I strongly recommend that the authors reword this statement to reflect our current understanding of the impact missing data can have on phylogenetic inference – and indeed put a relevant reference at the end. The current authors correctly cite Perreau and Tafforeau’s (2011) findings, but Perreau and Tafforeau didn’t broach the “reliable evolutionary inference” angle (which is why the current work is novel). For more papers on missing data, I recommend reading these reviews:

Wiens, J. J. (2003). Missing data, incomplete taxa, and phylogenetic accuracy.Systematic Biology, 52(4), 528538.

Wiens, J. J. (2006). Missing data and the design of phylogenetic analyses.Journal of biomedical informatics, 39(1), 3442.

Kearney, M., & Clark, J. M. (2003). Problems due to missing data in phylogenetic analyses including fossils: a critical review. Journal of Vertebrate Paleontology, 23(2), 263274.

3) “Arthropod fossils generally occur as adpressions…” or, indeed, mixtures of these (see e.g. Crato Formation).

4) “(E.g. pyritized in siderite nodules (Nitecki, 1979)).” While there often is a pyrite halo around siderite fossils, the fossils themselves are usually voids, and thus not pyritised see, e.g., any of my (Russell Garwood) Carboniferous arachnid/insect papers. I suggest you correct this (they are also pretty good to CT scan as a result). Some arthropod fossils are indeed pyritised though, I guess Beecher’s trilobite bed (which is also fairly 3D) or the Hunsruck Slate would both be good examples if you want some.

5) “However, in these fossil types, while internal and soft-tissue characters, such as eyes (Duncan and Briggs, 1996) and muscle fibres (Grimaldi, 2009) have been reported (Sutton, 2008), they have been rare and fragmentary.”

Again, this is something of a simplification – a range of soft-tissue, and indeed, internal characters, are known for whole animals (so they are not that fragmentary). They’re also not as rare as the authors suggest. To name some deposits which preserve internal characters (off the top of my head, so biased towards the Palaeozoic as that’s what I know better – but presumably this gets more, rather than less, frequent with younger rocks):

The Rhynie Cherts

Torridonian Cherts

Doushantuo Formation

Orsten

Gogo Formation

La Voulte-sur-Rhône

And all macerates from the Silurian/Devonian (e.g. Gilboa, Wenlock)

If you include just soft tissues, you can add many more, e.g.:

Chengjiang

All Burgess Shale-type deposits

All Ediacaran fossil-bearing deposits

The Silurian Lagerstatten

All Mazon-creek-type Lagerstatten

The Soom Shale

The Hunsrück Slates

Solnhofen Limestone

Crato Formation

I suggest the authors reword to highlight that soft tissue preservation, at least, is surprisingly common, and indeed, that this makes their work more important, as it will apply to a larger number of fossils.

6) Synchrotron X-ray microtomography has become established for detailed examination of fossils (Sutton, 2008). Microtomography has become well established enough in palaeontology that there is a book on it:

Sutton, M., Rahman, I., & Garwood, R. (2013). Techniques for virtual palaeontology. John Wiley & Sons.

The authors shouldn’t feel compelled to cite it, but there are a large number of appropriate references in it that may be of utility. I’d also encourage the authors to just say Microtomography, as palaeontologists have a tendency to run to a synchrotron for any scan they can do, where for the majority a decent lab-based scanner would do the job very well. This is something which doesn’t need to be further entrenched in the mindset.

7) “Recently especially for amber inclusions”

Not especially, as there are a comparable number of applications to fossil plants and seeds in a variety of preservational settings, Doushantuo fossils, macerated microfossils, and many more besides. I recommend the authors say “including fossils preserved in amber” or similar.

8) “Using the same experimental setup”

Why? Surely the beam energy and scanning conditions could (and should) be modified for each specimen?

Results and Discussion:

9) “It is mostly represented by air-filled spaces in the fossil”

What, the cuticle? So you have a void in the form of the organism? This isn’t 100% clear – I suggest you clarify. Furthermore, it’s quite hard to tell preserved cuticle from void in scans, as both will be significantly less dense than the surrounding rock. How can you be sure that features such as, for example, the accessory gland, are not still preserved cuticle.

10) “Large parts of alimentary canal and tracheal system are visible”

This is fantastic.

11) “While the middle part is apparently filled with mineral matrix but well-defined”

This is a bit hard to discern in the image – is this raw, unmodified data? If so, I suggest that adjusting the contrast to better display these features would be absolutely fine, and would make them clearer.

12) “Facilitated an extended description of the species according to modern taxonomic standards (Material and methods)”

Would this description not belong better in the Results, i.e. before this section? This would seem more logical to me.

13) “However, phylogenetic analysis of the more diverse character set now accessible separates these species widely”

This is clear. However, it is worth noting that a cladistic analysis could never show “a direct lineal relationship”. Rather this would be revealed as a sister group relationship between the two taxa, or – more likely – have intermedius as the earliest split in a clade including *striatus* – intermedius is only actually one node away from this, instead being in a polytomy that is sister to the *striatus* clade. Maybe you could clarify the above statement to reflect this.

14) “In all most parsimonious solutions *O. striatus* is nested within a group of mainly Nearctic species related to *O. flohri* Lewis, 1888.”

Based on what characters? And what characters is *O. intermedius* placed on the basis of? If these are not both internal then you could have got this without the CT work.

15) “Inclusion of internal characters of male genitalia for *O. intermedius* yields a better resolved consensus topology and a fuller view of the biogeographic and morphological history of the group.”

What does “better resolved” actually mean? Can you add an SI figure showing the trees of analysis with, and without, genitailic characters? Or with, but with those for *O. intermedius* coded is unknown in one and correctly coded in the other?

16) “Critical diagnostic differences in external morphology were also revealed by visualization of features previously obscured by matrix.”

What in particular?

This discussion of the phylogeny results could be strengthened by using including the above points. In addition, I had a fiddle with the data in TNT. To make the NEXUS load in this software, it needs to be simplified to:

#NEXUS

BEGIN DATA;

DIMENSIONS NTAX = 37 NCHAR=41;

FORMAT DATATYPE = STANDARD GAP = - MISSING = ? SYMBOLS = " 0 1 2 3 4";

MATRIX

'Cretonthophilus_tuberculatus' 12122143222?12211?312121?1?

'Anapleus_sp.' 1111211144121111113221212111111121211?221

'Peploglyptus_belfragei' 211121222411122111312221111111121?21112?

'Sculptura_kivuensis' 2212124321121221132121212122121?211211?

'Vuatuoxinus_borassicola' 2212222314121211113121212111111?

'Epiechinus_spDNA' 22112243111112211131222221221213221121111

'Epiechinus_rappi' 22122222241112211121212121111112221121112

'Onthophilus_intermedius' ?43211?1221132?121122222211?1211222

'Onthophilus_affinis' 212111211{2 4}2312211322112111111111212211211

'Onthophilus_alternatus' 22222143111212212321222222221222112111222

'Onthophilus_aonoi' 2221114323221222232?22?21111?2?111?1?

'Onthophilus_australis' 22112121141312211?31221111221111112?113?

'Onthophilus_cynomysi' 21212143211212111221221112111112122112322

'Onthophilus_deflectus' 22222143211312221332122111221211212211211

'Onthophilus_flavicornis' 22222143211?2221133?2121?222122?2?21212?

'Onthophilus_flohri' 22222143211312211332122122221221212111211

'Onthophilus_giganteus' 2221224211222221222?21222222?2222212211?

'Onthophilus_globulosus' 22211?432312122113?1?1?22?

'Onthophilus_intermixtus' 222221431113122113322221222212212?21112?

'Onthophilus_irregularis' 222221432313122223?1?22122?

'Onthophilus_julii' 222221432313122223321221?222122?

'Onthophilus_kirni' 21212222241222112121212122221122122112222

'Onthophilus_lecontei' 21212142111212111321211222111112122112222

'Onthophilus_lijiangensis' 2222114323121222232?21212221132?

'Onthophilus_niponensis' 222111432312122223222121?21111?1?111?1?

'Onthophilus_nodatus' 222221432112122123212212122212222111211?

'Onthophilus_ordinarius' 2?432122122223?2?1?2?2?211?2?

'Onthophilus_ostreatus' 22211143232212222322?22?222112?2?111?2?

'Onthophilus_pluricostatus' 222221431112122123212222222212222111213?

'Onthophilus_punctatus' 22211?432312122123?2?2221?2?1?22?

'Onthophilus_sculptilis' 2221114311121222232?222?22211221?1111312

'Onthophilus_silvae' 21222143221212111221112112?122?2?11211?

'Onthophilus_smetanai' 222111?1?1211132?22?1?21111?21112?1?

'Onthophilus_soltaui' 2121114211122211122?2122?221111?

'Onthophilus_striatus' 22222143211312211322122112221121112111211

'Onthophilus_thomomysi' 2121122214122211122?2122221111121?21122?2

'Onthophilus_wenzeli' 22222143111222111321222222221122112212322

END;

I was interested to see where *intermedius* came out using implied weighting (k=3), shown on the next page. The authors may like to include this in their Discussion, as it tells a slightly different story, and is fairly widely used in fossil arthropod phylogenies. I quote from one of my papers (Garwood and Dunlop, 2014) for more info: “Goloboff (1993) and Goloboff et al. (2008) provide an overview of this weighting scheme, whilst Legg, Sutton & Edgecombe (2013), Legg & Caron (2014) and Ortega-Hernández, Legg & Braddy (2013) provide justification of its use in a palaeontological context.”

Also, just to note, that we have found that viewing external morphology of fossils alone through CT: Garwood, R. J., & Dunlop, J. (2014). Three-dimensional reconstruction and the phylogeny of extinct chelicerate orders. PeerJ, 2, e641.

I would make the case that some of the changes you are seeing are down to just more data, or the inclusion of fossils themselves – as we found in the above exercise. To tie what you are seeing down to the inclusion of internal characters I would like to see a little more interrogation of the data as suggested above – obviously this paper is neither the time nor place to try and quantify this, or even say much in general, but there are relatively simple tests you can do through removal of characters and taxa to try and assess what is driving tree topology.

17) “Based on our examinations we can reconstruct the probable fossilization process of the Quercy *Onthophilus* specimens.”

I suggest you state here outright that some internals are replaced through mineralisation, and that some cuticle is preserved as void – as it is currently structured it takes quite a long time to get to this fact, which is useful for following the hypothesis for how this was preserved.

18) “By a liquid phosphoric solution”

What kind of solution – phosphate rich water?

19) “From a handful of localities”

If there are relatively few, I suggest you name them so the reader does not have to look them up.

20) “Including fragile appendices”

Do you mean appendages?

21) “Are circulating water”

Virtually any sediment will have circulating water in it. I’d reword to reflect this e.g. “Possible sources for high phosphorous concentrations in water circulating through the fissure fill are…”

22) “In contrast, an artificial ‘digital endocast’”

What you mean by this isn’t 100% clear to me.

23) “So far, the study of the anatomy of fossil arthropods has been largely confined to amber inclusions”

This is not an accurate statement. To choose outline just a limited number of examples of fossil arthropod anatomy in recent years, the Herefordshire group has published >15 papers using serial grinding to elucidate fossil anatomy. Here are three:

Sutton, M. D., Briggs, D. E., Siveter, D. J., Siveter, D. J., & Orr, P. J. (2002). The arthropod Offacolus kingi (Chelicerata) from the Silurian of Herefordshire, England: computer based morphological reconstructions and phylogenetic affinities. Proceedings of the Royal Society of London B: Biological Sciences, 269(1497), 1195-1203.

Briggs, D. E., Sutton, M. D., Siveter, D. J., & Siveter, D. J. (2004). A new phyllocarid (Crustacea: Malacostraca) from the Silurian fossil–Lagerstätte of Herefordshire, UK. Proceedings of the Royal Society of London B: Biological Sciences, 271(1535), 131-138.

Siveter, D. J., Briggs, D. E., Siveter, D. J., Sutton, M. D., & Joomun, S. C. (2013). A Silurian myodocope with preserved soft-parts: cautioning the interpretation of the shell-based ostracod record. Proceedings of the Royal Society of London B: Biological Sciences, 280(1752), 20122664.

Xiaoya Ma and colleagues have published at least four papers on fossil brain anatomy in arthropods:

Ma, X., Hou, X., Edgecombe, G. D., & Strausfeld, N. J. (2012). Complex brain and optic lobes in an early Cambrian arthropod. Nature, 490(7419), 258-261.

Tanaka, G., Hou, X., Ma, X., Edgecombe, G. D., & Strausfeld, N. J. (2013). Chelicerate neural ground pattern in a Cambrian great appendage arthropod.Nature, 502(7471), 364-367.

Cong, P., Ma, X., Hou, X., Edgecombe, G. D., & Strausfeld, N. J. (2014). Brain structure resolves the segmental affinity of anomalocaridid appendages.Nature.

Ma, X., Cong, P., Hou, X., Edgecombe, G. D., & Strausfeld, N. J. (2014). An exceptionally preserved arthropod cardiovascular system from the early Cambrian. Nature communications, 5.

I have published 18 papers since 2010 using CT to describe fossil arthropod anatomy from largely Carboniferous taxa. We’ve even had suggestions of internal anatomy and evo devo in trilobites:

Ortega-Hernández, J., & Brena, C. (2012). Ancestral patterning of tergite formation in a centipede suggests derived mode of trunk segmentation in trilobites. PloS one, 7(13), e52623.

None of which are in amber. The present authors need to rephrase this sentence to reflect the fact they are building on a significant body of work, with a deep history [see Pocock, R. I. (1911). monograph of the terrestrial Carboniferous Arachnida of Great Britain. for some truly excellent work on fossil arthropods anatomy]. There is still a representational bias, but it will be more complex than just arboreal, as it will be defined by taphonomic windows.

24) “May provide a highly complementary source of information on the evolutionary history of arthropods”

Yes, we’ve been saying this for a few years now – here’s a quote from a 2009 paper (Garwood et al 2009 Biology Letters):

“These results demonstrate the ability of XMT to differentiate the void left by the original organism’s decay within sideritic host material and the power of computer-based three-dimensional visualizations of the resultant datasets as a tool for morphological analysis.”

The current work shows the breadth of the technique, and further demonstrates its potential for widespread application. I feel the current authors’ point could be strengthened by citing some of the other 3D work on arthropods, and framed in the light of my comment above. The references don’t have to be my paper – see, for example:

Selden, P. A., Shear, W. A., & Sutton, M. D. (2008). Fossil evidence for the origin of spider spinnerets, and a proposed arachnid order. Proceedings of the National Academy of Sciences, 105(55), 20781-20785.

Bosselaers, J., Dierick, M., Cnudde, V., Masschaele, B., Van Hoorebeke, L., & Jacobs, P. (2010). High-resolution X-ray computed tomography of an extant new Donuea (Araneae: Liocranidae) species in Madagascan copal. Zootaxa, 2427(1), 25-35.)

Materials and methods:

25) “3D X-ray micro-computer tomography”

There is no such thing as computer tomography – I think this should read *computed* tomography.

26) “Parallel polychromatic X-ray beam, spectrally filtered”

How was the beam filtered? Did you use a filter material? Does the beamline have a variable-period undulator?

27) “The surface polygons were reduced”

Using what algorithm – quadric fidelity reduction? What was the original triangle count, and how much was it reduced by? This will give a useful indicator of just how much smoothing the process will have introduced, and is thus worth mentioning.

---

## [Author Response]

Summary:

*The authors show that the internal preservation of the Quercy fossils is unexpectedly excellent, and the authors reveal this with technical skill. The level of preservation is incredible for a non-amber fossil. The preservation of soft anatomy such as an insect's tracheal system makes the study just the kind of thing that should have a large impact in palaeontology. Further, there is actually more behind this manuscript than a technical/techniques paper, and in fact the phylogenetic finding that the Quercy fossil histerid species is most closely related to a North American species is interesting (should mention this in the Abstract). Surely, this biogeographic finding would not have been possible without such remarkable preservation. Regarding the mode of preservation, it is also quite likely that internal decomposition of the original insect was quickly halted because of the relatively antiseptic nature of the phosphate-saturated water that infiltrated the body.*

*O. intermedius* was not placed within a group of North American species, but maybe sister to the Far-Eastern *O. silvae*. The text accurately describes the placement (and uncertainty), especially with the revisions that discuss slightly better resolution based on reweighted characters.

As this is a rather complex discussion, we decided not to mention the detailed results of the phylogenetic analysis in the Abstract.

*Essential revisions:*

One of the reviewers (Russel Garwood) provides direct comments on text passages that should be responded to (appended below). Several of them refer to providing more citations for the general technical background of the study.

Please clarify the following sentences in light of the accompanying comments:

Abstract:

1) “Internal characters in fossil arthropods are crucial to establishing systematic position, ecological role and evolutionary trends.”

I believe this is an oversimplification of a complex issue – see comments below.

We now mention both external and internal characters and extended the Introduction (see comments below).

Introduction:

2) “Internal characters are crucial for assessing the systematic placement and ecological role of organisms and provide essential information for reliable evolutionary inference. In many arthropods, they include critical systematic characters (Perreau and Tafforeau, 2011).”

This statement is a simplification of what is in fact a complex and under-researched area. Internal characters in fossils are just a form of missing data, and are no more – or less – important, than external data. I suggest the authors check out the following paper: Sansom, R. S. (2014). Bias and sensitivity in the placement of fossil taxa resulting from interpretations of missing data. Systematic biology, syu093.

It’s relatively safe to say that more data is useful in a cladistic phylogenetic context, but (as Sansom 2015 demonstrates) – what is more important when dealing with missing data is to correctly code data as such, rather than introducing false absences (which is the biggest risk when dealing with fossils). Nobody has, to my knowledge, published any work on whether the loss of internal characters leads to biases in phylogenies. This is where it would be a significant issue. I strongly recommend that the authors reword this statement to reflect our current understanding of the impact missing data can have on phylogenetic inference – and indeed put a relevant reference at the end. The current authors correctly cite Perreau and Tafforeau’s (2011) findings, but Perreau and Tafforeau didn’t broach the “reliable evolutionary inference” angle (which is why the current work is novel). For more papers on missing data, I recommend reading these reviews:

*Wiens, J. J. (2003). Missing data, incomplete taxa, and phylogenetic accuracy.Systematic Biology, 52(4), 528538.*

Wiens, J. J. (2006). Missing data and the design of phylogenetic analyses.Journal of biomedical informatics, 39(1), 3442.

Kearney, M., & Clark, J. M. (2003). Problems due to missing data in phylogenetic analyses including fossils: a critical review. Journal of Vertebrate Paleontology, 23(2), 263274.

We followed the suggestion and reworded the respective section to address the problem of missing data.

3) “Arthropod fossils generally occur as adpressions…” or, indeed, mixtures of these (see e.g. Crato Formation).

We now mention the occurrence of mixtures.

4) “(E.g. pyritized in siderite nodules (Nitecki, 1979)).” While there often is a pyrite halo around siderite fossils, the fossils themselves are usually voids, and thus not pyritised see, e.g., any of my (Russell Garwood) Carboniferous arachnid/insect papers. I suggest you correct this (they are also pretty good to CT scan as a result). Some arthropod fossils are indeed pyritised though, I guess Beecher’s trilobite bed (which is also fairly 3D) or the Hunsruck Slate would both be good examples if you want some.

Thank you for the suggestion. We changed the sentence and added a new reference (Garwood et al. 2009). To avoid confusion, we also exchanged “hollow casts” by “calcareous incrustations” in the following sentence.

5) “However, in these fossil types, while internal and soft-tissue characters, such as eyes (Duncan and Briggs, 1996) and muscle fibres (Grimaldi, 2009) have been reported (Sutton, 2008), they have been rare and fragmentary.”

Again, this is something of a simplification – a range of soft-tissue, and indeed, internal characters, are known for whole animals (so they are not that fragmentary). They’re also not as rare as the authors suggest. To name some deposits which preserve internal characters (off the top of my head, so biased towards the Palaeozoic as that’s what I know better – but presumably this gets more, rather than less, frequent with younger rocks):

The Rhynie Cherts

Torridonian Cherts

Doushantuo Formation

Orsten

Gogo Formation

La Voulte-sur-Rhône

And all macerates from the Silurian/Devonian (e.g. Gilboa, Wenlock)

If you include just soft tissues, you can add many more, e.g.:

Chengjiang

All Burgess Shale-type deposits

All Ediacaran fossil-bearing deposits

The Silurian Lagerstatten

All Mazon-creek-type Lagerstatten

The Soom Shale

The Hunsrück Slates

Solnhofen Limestone

Crato Formation

I suggest the authors reword to highlight that soft tissue preservation, at least, is surprisingly common, and indeed, that this makes their work more important, as it will apply to a larger number of fossils.

Indeed we underestimated the fossil record of internal preservation in non-amber arthropods. We rephrased the section accordingly and added several new references about soft tissue preservation from Chenjiang-Formation, Burges-Shale, Katian Stage Lorraine Group, and Herefordshire.

6) Synchrotron X-ray microtomography has become established for detailed examination of fossils (Sutton, 2008). Microtomography has become well established enough in palaeontology that there is a book on it:

Sutton, M., Rahman, I., & Garwood, R. (2013). Techniques for virtual palaeontology. John Wiley & Sons.

The authors shouldn’t feel compelled to cite it, but there are a large number of appropriate references in it that may be of utility. I’d also encourage the authors to just say Microtomography, as palaeontologists have a tendency to run to a synchrotron for any scan they can do, where for the majority a decent lab-based scanner would do the job very well. This is something which doesn’t need to be further entrenched in the mindset.

This is right, of course. We followed the advice, changed the term accordingly and added several references.

7) “Recently especially for amber inclusions”

Not especially, as there are a comparable number of applications to fossil plants and seeds in a variety of preservational settings, Doushantuo fossils, macerated microfossils, and many more besides. I recommend the authors say “including fossils preserved in amber” or similar.

We followed the suggestion.

8) “Using the same experimental setup”

Why? Surely the beam energy and scanning conditions could (and should) be modified for each specimen?

It depends on the scientific question. If the aim is to achieve best possible feature contrast for a single specimen, then experiment conditions like the energy spectrum should be optimized for each scan. In our case, however, the aim was to have the same scan conditions to facilitate a direct comparison between the extant and fossil specimen. We therefore made a joint optimization of the experimental conditions with both specimens, such that the feature contrast was satisfactory for both samples.

Results and Discussion:

9) “It is mostly represented by air-filled spaces in the fossil”

What, the cuticle? So you have a void in the form of the organism? This isn’t 100% clear – I suggest you clarify. Furthermore, it’s quite hard to tell preserved cuticle from void in scans, as both will be significantly less dense than the surrounding rock. How can you be sure that features such as, for example, the accessory gland, are not still preserved cuticle.

The gray value outside the sample serves as an in-dataset reference for air, and we see the same gray value range in the parts of the animal were cuticle should be.

*10) “Large parts of alimentary canal and tracheal system are visible”*

This is fantastic.

Yes, it is. Thank you.

11) “While the middle part is apparently filled with mineral matrix but well-defined”

This is a bit hard to discern in the image – is this raw, unmodified data? If so, I suggest that adjusting the contrast to better display these features would be absolutely fine, and would make them clearer.

Yes, the figure shows sliced of unmodified reconstructed data. We followed the suggestion and adjusted the contrast of the figure.

12) “Facilitated an extended description of the species according to modern taxonomic standards (Material and methods)”

Would this description not belong better in the Results, i.e. before this section? This would seem more logical to me.

In our opinion the species description stands on rather its own and aims particularly at taxonomists. The formal language would clash with the style of the Results and Discussion section and impede the reading flow for the non-specialist. Since the appendices are included in the PDF (see e.g. http://elifesciences.org/content/4/e05856/), taxonomists would find this important information prominently emphasized at the end of the main text.

13) “However, phylogenetic analysis of the more diverse character set now accessible separates these species widely”

This is clear. However, it is worth noting that a cladistic analysis could never show “a direct lineal relationship”. Rather this would be revealed as a sister group relationship between the two taxa, or – more likely – have intermedius as the earliest split in a clade including striatus – intermedius is only actually one node away from this, instead being in a polytomy that is sister to the striatus clade. Maybe you could clarify the above statement to reflect this.

We have revised the text to more clearly relate Handschin’s wording and how we translate it into a testable hypothesis.

14) “In all most parsimonious solutions O. striatus is nested within a group of mainly Nearctic species related to O. flohri Lewis, 1888.”

Based on what characters? And what characters is O. intermedius placed on the basis of? If these are not both internal then you could have got this without the CT work.

Mapping of specific character changes on a new Tree Figure (Figure 5) now better illustrates what classes of characters support the critical branches. Indeed internal characters provide critical synapomorphies.

15) “Inclusion of internal characters of male genitalia for O. intermedius yields a better resolved consensus topology and a fuller view of the biogeographic and morphological history of the group.”

What does “better resolved” actually mean? Can you add an SI figure showing the trees of analysis with, and without, genitailic characters? Or with, but with those for O. intermedius coded is unknown in one and correctly coded in the other?

We reworded this section to reflect the fact that while internal characters did provide important synapomorphies, the accurate assessment of external characters (which were hidden by matrix) provided by microtomography was also essential to the resolution we found. We did some analyses with genitalic characters of *O. intermedius* missing to test their effects. In fact the same topology was obtained, but the character reconstructions were different, and the larger clade was defined by external characters with lower CIs. Inclusion of genitalic characters doesn’t affect the clades discovered, just how they are defined. The new text clarifies this subtler point.

16) “Critical diagnostic differences in external morphology were also revealed by visualization of features previously obscured by matrix.”

What in particular?

Examples of important external characters are now given.

This discussion of the phylogeny results could be strengthened by using including the above points. In addition, I had a fiddle with the data in TNT. To make the NEXUS load in this software, it needs to be simplified to:

#NEXUS

BEGIN DATA;

DIMENSIONS NTAX = 37 NCHAR=41;

FORMAT DATATYPE = STANDARD GAP = - MISSING = ? SYMBOLS = " 0 1 2 3 4";

MATRIX

'Cretonthophilus_tuberculatus' 12122143222?12211?312121?1?

'Anapleus_sp.' 1111211144121111113221212111111121211?221

'Peploglyptus_belfragei' 211121222411122111312221111111121?21112?

'Sculptura_kivuensis' 2212124321121221132121212122121?211211?

'Vuatuoxinus_borassicola' 2212222314121211113121212111111?

'Epiechinus_spDNA' 22112243111112211131222221221213221121111

'Epiechinus_rappi' 22122222241112211121212121111112221121112

'Onthophilus_intermedius' ?43211?1221132?121122222211?1211222

'Onthophilus_affinis' 212111211{2 4}2312211322112111111111212211211

'Onthophilus_alternatus' 22222143111212212321222222221222112111222

'Onthophilus_aonoi' 2221114323221222232?22?21111?2?111?1?

'Onthophilus_australis' 22112121141312211?31221111221111112?113?

'Onthophilus_cynomysi' 21212143211212111221221112111112122112322

'Onthophilus_deflectus' 22222143211312221332122111221211212211211

'Onthophilus_flavicornis' 22222143211?2221133?2121?222122?2?21212?

'Onthophilus_flohri' 22222143211312211332122122221221212111211

'Onthophilus_giganteus' 2221224211222221222?21222222?2222212211?

'Onthophilus_globulosus' 22211?432312122113?1?1?22?

'Onthophilus_intermixtus' 222221431113122113322221222212212?21112?

'Onthophilus_irregularis' 222221432313122223?1?22122?

'Onthophilus_julii' 222221432313122223321221?222122?

'Onthophilus_kirni' 21212222241222112121212122221122122112222

'Onthophilus_lecontei' 21212142111212111321211222111112122112222

'Onthophilus_lijiangensis' 2222114323121222232?21212221132?

'Onthophilus_niponensis' 222111432312122223222121?21111?1?111?1?

'Onthophilus_nodatus' 222221432112122123212212122212222111211?

'Onthophilus_ordinarius' 2?432122122223?2?1?2?2?211?2?

'Onthophilus_ostreatus' 22211143232212222322?22?222112?2?111?2?

'Onthophilus_pluricostatus' 222221431112122123212222222212222111213?

'Onthophilus_punctatus' 22211?432312122123?2?2221?2?1?22?

'Onthophilus_sculptilis' 2221114311121222232?222?22211221?1111312

'Onthophilus_silvae' 21222143221212111221112112?122?2?11211?

'Onthophilus_smetanai' 222111?1?1211132?22?1?21111?21112?1?

'Onthophilus_soltaui' 2121114211122211122?2122?221111?

'Onthophilus_striatus' 22222143211312211322122112221121112111211

'Onthophilus_thomomysi' 2121122214122211122?2122221111121?21122?2

'Onthophilus_wenzeli' 22222143111222111321222222221122112212322

END;

We did change the NEXUS file to use more generic (non-Mesquite) headers.

I was interested to see where intermedius came out using implied weighting (k=3), shown on the next page. The authors may like to include this in their Discussion, as it tells a slightly different story, and is fairly widely used in fossil arthropod phylogenies. I quote from one of my papers (Garwood and Dunlop, 2014) for more info: “Goloboff (1993) and Goloboff et al. (2008) provide an overview of this weighting scheme, whilst Legg, Sutton & Edgecombe (2013), Legg & Caron (2014) and Ortega-Hernández, Legg & Braddy (2013) provide justification of its use in a palaeontological context.”

Also, just to note, that we have found that viewing external morphology of fossils alone through CT: Garwood, R. J., & Dunlop, J. (2014). Three-dimensional reconstruction and the phylogeny of extinct chelicerate orders. PeerJ, 2, e641.

I would make the case that some of the changes you are seeing are down to just more data, or the inclusion of fossils themselves – as we found in the above exercise. To tie what you are seeing down to the inclusion of internal characters I would like to see a little more interrogation of the data as suggested above – obviously this paper is neither the time nor place to try and quantify this, or even say much in general, but there are relatively simple tests you can do through removal of characters and taxa to try and assess what is driving tree topology.

We now discuss reweighted trees in the text. See also our comments above.

17) “Based on our examinations we can reconstruct the probable fossilization process of the Quercy Onthophilus specimens.”

I suggest you state here outright that some internals are replaced through mineralisation, and that some cuticle is preserved as void – as it is currently structured it takes quite a long time to get to this fact, which is useful for following the hypothesis for how this was preserved.

We followed the suggestion and extended the statement.

18) “By a liquid phosphoric solution”

What kind of solution – phosphate rich water?

Yes, we changed the term.

19) “From a handful of localities”

If there are relatively few, I suggest you name them so the reader does not have to look them up.

We followed the suggestion.

20) “Including fragile appendices”

Do you mean appendages?

Yes, corrected.

21) “Are circulating water”

Virtually any sediment will have circulating water in it. I’d reword to reflect this e.g. “Possible sources for high phosphorous concentrations in water circulating through the fissure fill are…”

22) “In contrast, an artificial ‘digital endocast’”

What you mean by this isn’t 100% clear to me.

We added a more detailed description to the Material and methods.

23) “So far, the study of the anatomy of fossil arthropods has been largely confined to amber inclusions”

This is not an accurate statement. To choose outline just a limited number of examples of fossil arthropod anatomy in recent years, the Herefordshire group has published >15 papers using serial grinding to elucidate fossil anatomy. Here are three:

Sutton, M. D., Briggs, D. E., Siveter, D. J., Siveter, D. J., & Orr, P. J. (2002). The arthropod Offacolus kingi (Chelicerata) from the Silurian of Herefordshire, England: computer based morphological reconstructions and phylogenetic affinities. Proceedings of the Royal Society of London B: Biological Sciences, 269(1497), 1195-1203.

Briggs, D. E., Sutton, M. D., Siveter, D. J., & Siveter, D. J. (2004). A new phyllocarid (Crustacea: Malacostraca) from the Silurian fossil–Lagerstätte of Herefordshire, UK. Proceedings of the Royal Society of London B: Biological Sciences, 271(1535), 131-138.

Siveter, D. J., Briggs, D. E., Siveter, D. J., Sutton, M. D., & Joomun, S. C. (2013). A Silurian myodocope with preserved soft-parts: cautioning the interpretation of the shell-based ostracod record. Proceedings of the Royal Society of London B: Biological Sciences, 280(1752), 20122664.

Xiaoya Ma and colleagues have published at least four papers on fossil brain anatomy in arthropods:

Ma, X., Hou, X., Edgecombe, G. D., & Strausfeld, N. J. (2012). Complex brain and optic lobes in an early Cambrian arthropod. Nature, 490(7419), 258-261.

Tanaka, G., Hou, X., Ma, X., Edgecombe, G. D., & Strausfeld, N. J. (2013). Chelicerate neural ground pattern in a Cambrian great appendage arthropod.Nature, 502(7471), 364-367.

Cong, P., Ma, X., Hou, X., Edgecombe, G. D., & Strausfeld, N. J. (2014). Brain structure resolves the segmental affinity of anomalocaridid appendages.Nature.

Ma, X., Cong, P., Hou, X., Edgecombe, G. D., & Strausfeld, N. J. (2014). An exceptionally preserved arthropod cardiovascular system from the early Cambrian. Nature communications, 5.

I have published 18 papers since 2010 using CT to describe fossil arthropod anatomy from largely Carboniferous taxa. We’ve even had suggestions of internal anatomy and evo devo in trilobites:

Ortega-Hernández, J., & Brena, C. (2012). Ancestral patterning of tergite formation in a centipede suggests derived mode of trunk segmentation in trilobites. PloS one, 7(13), e52623.

None of which are in amber. The present authors need to rephrase this sentence to reflect the fact they are building on a significant body of work, with a deep history [see Pocock, R. I. (1911). monograph of the terrestrial Carboniferous Arachnida of Great Britain. for some truly excellent work on fossil arthropods anatomy]. There is still a representational bias, but it will be more complex than just arboreal, as it will be defined by taphonomic windows.

We followed the suggestions and rephrased this paragraph to better reflect the fossil biases. This should better point out the position and value of the Quercy arthropods. Several new references focusing on non-amber arthropod fossils were added to the Introduction (see also comment regarding internal preservation in non-amber arthropods).

24) “May provide a highly complementary source of information on the evolutionary history of arthropods”

Yes, we’ve been saying this for a few years now – here’s a quote from a 2009 paper (Garwood et al 2009 Biology Letters):

“These results demonstrate the ability of XMT to differentiate the void left by the original organism’s decay within sideritic host material and the power of computer-based three-dimensional visualizations of the resultant datasets as a tool for morphological analysis.”

The current work shows the breadth of the technique, and further demonstrates its potential for widespread application. I feel the current authors’ point could be strengthened by citing some of the other 3D work on arthropods, and framed in the light of my comment above. The references don’t have to be my paper – see, for example:

Selden, P. A., Shear, W. A., & Sutton, M. D. (2008). Fossil evidence for the origin of spider spinnerets, and a proposed arachnid order. Proceedings of the National Academy of Sciences, 105(55), 20781-20785.

Bosselaers, J., Dierick, M., Cnudde, V., Masschaele, B., Van Hoorebeke, L., & Jacobs, P. (2010). High-resolution X-ray computed tomography of an extant new Donuea (Araneae: Liocranidae) species in Madagascan copal. Zootaxa, 2427(1), 25-35.)

We followed the advice and added several new references featuring microtomography of arthropods (also of extant specimens) in the Introduction.

Materials and methods:

25) “3D X-ray micro-computer tomography”

There is no such thing as computer tomography – I think this should read computed tomography.

Yes, corrected.

26) “Parallel polychromatic X-ray beam, spectrally filtered”

How was the beam filtered? Did you use a filter material? Does the beamline have a variable-period undulator?

We extended the section with the requested information.

27) “The surface polygons were reduced”

Using what algorithm – quadric fidelity reduction? What was the original triangle count, and how much was it reduced by? This will give a useful indicator of just how much smoothing the process will have introduced, and is thus worth mentioning.

We agree that this information is worth mentioning, but don’t find it necessary to list the polygon count of all reconstructions done for this study. Therefore we now exemplarily mention the polygon numbers for the most important specimen F1994. The raw mesh of the data contained ca. 30 million polygons, the reduced version 3 million and the PDF version ca. 800,000. CINEMA 4D was used for polygon reduction. Unfortunately, the algorithm is not revealed by the software.